# An Algorithm for Choosing the Optimal Number of Muscle Synergies during Walking

**DOI:** 10.3390/s21103311

**Published:** 2021-05-11

**Authors:** Riccardo Ballarini, Marco Ghislieri, Marco Knaflitz, Valentina Agostini

**Affiliations:** 1Department of Electronics and Telecommunications, Politecnico di Torino, 10129 Turin, Italy; riccardo.ballarini@studenti.polito.it (R.B.); marco.ghislieri@polito.it (M.G.); marco.knaflitz@polito.it (M.K.); 2PoliTo^BIO^Med Lab, Politecnico di Torino, 10129 Turin, Italy

**Keywords:** gait, locomotion, motor module, number of synergies, VAF

## Abstract

In motor control studies, the 90% thresholding of variance accounted for (VAF) is the classical way of selecting the number of muscle synergies expressed during a motor task. However, the adoption of an arbitrary cut-off has evident drawbacks. The aim of this work is to describe and validate an algorithm for choosing the optimal number of muscle synergies (ChoOSyn), which can overcome the limitations of VAF-based methods. The proposed algorithm is built considering the following principles: (1) muscle synergies should be highly consistent during the various motor task epochs (i.e., remaining stable in time), (2) muscle synergies should constitute a base with low intra-level similarity (i.e., to obtain information-rich synergies, avoiding redundancy). The algorithm performances were evaluated against traditional approaches (threshold-VAF at 90% and 95%, elbow-VAF and plateau-VAF), using both a simulated dataset and a real dataset of 20 subjects. The performance evaluation was carried out by analyzing muscle synergies extracted from surface electromyographic (sEMG) signals collected during walking tasks lasting 5 min. On the simulated dataset, ChoOSyn showed comparable performances compared to VAF-based methods, while, in the real dataset, it clearly outperformed the other methods, in terms of the fraction of correct classifications, mean error (ME), and root mean square error (RMSE). The proposed approach may be beneficial to standardize the selection of the number of muscle synergies between different research laboratories, independent of arbitrary thresholds.

## 1. Introduction

Muscle synergies are a valuable tool to understand the mechanisms behind motor control in a quantitative and non-invasive way. Applications range from the medical field (e.g., monitoring of patients suffering from neurological/neurodegenerative diseases [1,2,3] or joint disorders [4]), to the rehabilitation field (e.g., pre/post-treatment comparisons [5,6]), to the robotic field (e.g., control of robotic devices or exoskeletons [7,8]), to the sport field [9].

The hypothesis of muscle synergies provides an insight into how the central nervous system (CNS) is able to manage a highly complex system with many muscles and joints. Indeed, the basis of this hypothesis is the ability of the CNS to reduce a large number of degrees of freedom of the movement thanks to the combination of a few discrete elements [10]. In other words, to generate movements, the CNS would not control the different muscles individually, but through functional groups, called muscle synergies.

Muscle synergies are usually extracted from surface electromyography (sEMG) signals, properly pre-processed, using the non-negative matrix factorization (NMF) algorithm [11,12]. This factorization algorithm requires the number of muscle synergies (*n*) as an input, which is not known a priori. Therefore, the factorization is typically run several times, considering different numbers of synergies (*n_i_* = [*n_min_*, *n_max_*]). The only constraint is that the number of synergies must not exceed the number of muscles (*m*) considered in the sEMG acquisition (*n_max_* ≤ *m*); otherwise, the meaning of “synergy” itself would be lost. Afterward, in post-processing, one has to choose the “correct” number of synergies *n_c_* (with *n_min_* ≤ *n_c_* ≤ *n_max_*), i.e., the input that feeds the factorization algorithm providing “good” results, with a “small-enough” reconstruction error. In recent years, the correct number of muscle synergies (*n_c_*) has been proposed as a meaningful feature for the analysis of motor control strategies in pathological populations [13,14,15,16,17]. A decreased neuromuscular complexity during gait has been assessed in post-stroke patients with respect to a healthy population [13]. Similar results were also found in another work [14], in which a reduced number of muscle synergies (two to four muscle synergies) were observed in the affected side of post-acute stroke patients with respect to a healthy population (four muscle synergies) while executing cycling training. These studies suggest that the number of muscle synergies and their composition could be correlated with motor control capacity and its reduction in pathological conditions [13,14,15,16,17].

Here lies one of the main issues of the muscle synergy extraction process: currently there is a lack of reliable methodologies for choosing the optimal number of muscle synergies. Most of the published studies choose *n_c_* based on the reconstruction accuracy of the factorization, through the variance accounted for (VAF) [1,2,18,19,20,21,22,23,24,25]. To a lesser extent, the coefficient of determination *R^2^* [16,26,27] is also used, which is not conceptually different from VAF. However, this approach requires the selection of an arbitrary threshold for the VAF. The number *n_c_* is defined as the smallest number of synergies that ensures a VAF value above the threshold. In literature, the VAF threshold is commonly set at 90% [1,2,4,7,18,20,21,22,23,24] and less frequently at 95% [19,25]. This method is very simple to implement, but it has several drawbacks: the threshold is arbitrary, it is set without an objective motivation, and there is not a single threshold value shared by all researchers. A few works have explored alternatives to VAF-based criteria. In particular, a statistical approach uses unstructured sEMG signals generated by randomly shuffling the original data across time and muscle [16], while other works consider the variability of muscle synergies between task cycles [28], or a task decoding-based metric [29,30].

The aim of this work is to overcome VAF-based methods using a data-driven approach. We designed and validated an algorithm for choosing the optimal number of muscle synergies (ChoOSyn), based on two parameters directly extracted from muscle synergies during locomotion: (1) the consistency within the motor task epochs (to identify synergies that are stable over the duration of the walking task), (2) the intra-level dissimilarity between synergies (to identify a base of information-rich synergies, avoiding unnecessary redundancy). Both a simulated and a real dataset were used to compare the performance of ChoOSyn against VAF-based methods.

## 2. Materials and Methods

### 2.1. Real Dataset

The real dataset originates from the retrospective analysis of sEMG signals previously recorded at Biolab (Politecnico di Torino, Italy) during gait analysis sessions [22,23,24]. The dataset contains gait signals from 20 healthy adults: 9 males (age: 56.9 ± 9.8 years, height: 1.71 ± 0.10 m, weight: 79.1 ± 22.0 kg) and 11 females (age: 51.5 ± 10.1 years, height: 1.66 ± 0.09 m, weight: 74.5 ± 24.0 kg).

Subjects walked at a self-selected speed for approximately 5 min. SEMG signals were acquired using the multi-channel recording system STEP32 for Statistical Gait Analysis (Medical Technology, Turin, Italy) [31]. The electrodes were positioned over the following 12 muscles of the dominant lower limb and over the trunk (bilaterally): Gluteus Medius (GMD), Tensor Fasciae Latae (TFL), Rectus Femoris (RF), Vastus Medialis (VM), Lateral Hamstring (LH), Medial Hamstring (MH), Lateral Gastrocnemius (LGS), Peroneus Longus (PL), Soleus (SOL), Tibialis Anterior (TA), and both right and left Longissimus Dorsii (LD_R_ and LD_L_) muscles.

The volunteers enrolled in this work signed a written informed consent to participate in a study concerning muscle synergies adopted by healthy subjects during locomotion. The experimental protocol conformed to the principles of the Helsinki declaration.

### 2.2. Simulated Dataset

Similar to previous studies [21,32,33], pseudo-real sEMG data have been generated from the real dataset to simulate the muscle activity during gait (simulated dataset). The following steps were used to generate simulated data:From the dataset of 20 subjects, 15 subjects were extracted, showing *n* = 4 (5 subjects), *n* = 5 (5 subjects), and *n* = 6 (5 subjects) clearly recognizable muscle synergies, as assessed by expert operators (V.A. and M.G.). Hence, for each subject, activation coefficients (C) and weight vectors (W) were obtained. Figure 1A shows an example of muscle synergies (*n* = 5) representative of a specific subject.For each group of 5 subjects, data augmentation was performed to obtain 25 “simulated subjects”, considering all the possible combinations of W and C. In other words, the matrix of weight vectors of the first subject ($W_{\mathrm{subj}1}$) was combined with the coefficient matrix of every subject in the group ($W_{\mathrm{subj}1}$$\;C_{\mathrm{subj}1}$, $W_{\mathrm{subj}1}$$\;C_{\mathrm{subj}2}$, … $W_{\mathrm{subj}1}$$\;C_{\mathrm{subj}5}$), and the same was performed for the other weight matrixes ($W_{\mathrm{subj}2}$, … $W_{\mathrm{subj}5}$), obtaining 25 sets of muscle synergies. Overall, 25 sets were obtained with *n* = 4, 25 sets with *n* = 5, and 25 sets with *n* = 6, for a total of 75 sets.For each set of W and C, each muscle’s envelope was reconstructed as the product W
_muscle_ * C, where W
_muscle_ is the weight vector of a specific muscle. Figure 1B provides an example for the LGS muscle.For each muscle’s envelope, a simulated sEMG signal (S) was generated by multiplying the envelope by a zero-mean Gaussian process (G_S_) with standard deviation σ = 1 a.u. (Figure 1C). At this step, no additive noise was superimposed on the signals. This does not mean that there was “no noise”, but rather that additional noise to the noise originally present in the envelope was not introduced.Then, different levels of background noise were added to obtain different SNR values (15 dB, 20 dB, 25 dB, and 30 dB), through a zero-mean Gaussian process (G_N_) with a standard deviation $\sigma =1/10^{SNR/20}$ a.u. [21,34]. Figure 1D shows an example in which SNR was equal to 20 dB. The formula below (1) summarizes how each simulated sEMG signal was generated:


(1)$$S=W_{\mathrm{muscle}}\;\times \;C\;\times \;G_{S}\;+\;G_{N}$$

Therefore, a total of 375 simulated sets were obtained, since we introduced both signals with no additive noise (75 sets) and signals with 4 different SNR values (75 × 4 sets).

### 2.3. Muscle Synergy Extraction and Sorting

After sEMG pre-processing [21,22,23,24], muscle synergies were extracted and properly ordered as outlined in Figure 2.

First, gait cycles were segmented and time-normalized to 1000 samples. Second, signals were high-pass filtered at 35 Hz through an 8th order Butterworth digital filter to attenuate slow movement artifacts and baseline wandering [22,35,36]. Third, signals were demeaned and rectified. Fourth, rectified signals were low-pass filtered at 12 Hz through a 5th order Butterworth digital filter to obtain the sEMG envelopes [22,35,36]. Fifth, each envelope was normalized in amplitude with respect to its global maximum. Then, we concatenated 10 adjacent gait cycles [37]. If the walk contained *N* gait cycles, the total number of subgroups was calculated rounding down *N*/10 to the smallest integer. As an example, if the walk contained 152 gait cycles, we considered 15 subgroups. Afterward, muscle synergies were extracted for each 10-cycle subgroup [21,22,23,24] through non-negative matrix factorization (NMF) [11,12]. The NMF algorithm models the original sEMG data as a linear combination of weight vectors (W) and activation coefficients (C), whose dimensions depend on the selected number of muscle synergies. In particular, the former models the time-independent contribution of each muscle to a specific muscle synergy, while the latter describes the time-dependent modulation of each muscle synergy. Instead of using the multiplicative update rule of the standard NMF approach [12], we chose to use another version of the algorithm, NMF with alternating non-negative least-squares (NMF/ANLS) [38], due to its advantages in terms of reduced computational time [38]. For the NMF/ANLS we set the following parameters: maximum iterations = 1000 [22], reruns = 5, residual error <10^−6^ [22], and output variation <10^−6^ [22]. To explore different solutions, the NMF algorithm was run several times on the same original sEMG data by changing the number of muscle synergies *n* in the range [1,8].

Finally, the 10-gait-cycle activation coefficient (10,000 samples) was time-averaged across windows of 1000 samples. Thus, we obtained an average activation coefficient for each subgroup.

The factorization returns W (and C) in a different order for each subgroup, and, therefore, proper sorting was required to average the correspondent W (and C) between the subgroups. For each number of synergies (*n*), we applied a *k*-means algorithm to reorder the weight vectors across the different subgroups (number of clusters: *n*, distance metric: cosine similarity, max iterations: 10^5^, replicas: 15) [22]. Activation coefficients were then sorted accordingly.

### 2.4. Choosing the Optimal Number of Synergies (ChoOSyn)

The algorithm for choosing the optimal number of muscle synergies (ChoOSyn algorithm) is based on the two following muscle synergy features:High consistency across time [22,23], that supports the possibility of finding a solution that is as stable as possible among the various 10-cycle subgroups of the motor taskLow similarity across synergies, to avoid selecting muscle synergies containing redundant information.

We chose these criteria after considering the characteristics of muscle synergies extracted from the real dataset.

In the following sections, we introduce the mathematical description of the parameters used to quantify the features described above. These parameters are a function of the number of synergies, so they assume a specific value for each number of synergies. They are also applicable for *n* ≥ 2, because the similarity parameter cannot be extracted at *n* = 1 (since there is only one synergy).

#### 2.4.1. Intra-Cluster Variability

The intra-cluster variability (ICV) quantifies the possible inconsistency of weight vectors ($ICV_{W}$) and activation coefficients ($ICV_{C}$) across time, i.e., between the different subgroups of 10 gait cycles. Its purpose is to quantify the level of variability of a given synergy during the considered task.

More specifically, for each number of synergies *n* (2 ≤ *n* ≤ 8), for each synergy *i* (with *i* = 1, …, *n*), and for each subgroup *j*, we calculate the distance between each “cluster element” ($W^{ij}$ and $C^{ij}$) and the “cluster centroid” (${\bar{W}}^{i}$ and ${\bar{C}}^{i}$), through cosine similarity [22,23,39]. Then, ICV is defined as:(2)$$ICV_{W}=max\left(\bar{1-\frac{W^{ij}\;\cdot \;{\bar{W}}^{i}}{\left\|W^{ij}\|\;\|{\bar{W}}^{i}\right\|}}\right)$$
(3)$$ICV_{C}=max\left(\bar{1-\frac{C^{ij}\;\cdot \;{\bar{C}}^{i}}{\left\|C^{ij}\|\;\|{\bar{C}}^{i}\right\|}}\right)$$
for the weights and the coefficients, respectively. Notice that the average operator is always applied across subgroups [22,23]. The “max” function is used to select the most variable muscle synergy (“worst” condition), obtaining a single ICV value for each *n* value. The ICV value ranges from 0 (i.e., perfectly repeatable muscle synergy between the different subgroups) to 1 (i.e., completely different muscle synergy across subgroups).

#### 2.4.2. Weight Similarity

The parameter weight similarity (WS) is introduced to select the two most similar weight vectors (“worst-case”) belonging to different muscle synergies.

For each number of synergies *n* (2 ≤ *n* ≤ 8), and for each synergy *i* (with *i* = 1, …, *n*), the average weight vector across subgroups is considered (${\bar{W}}^{i}$), representing the weights of a specific synergy over the entire locomotion task. Then, “cosine similarity” is introduced to quantify the degree of correlation between each couple of weight vectors $\left({\bar{W}}^{i}1\;{\bar{W}}^{k}\right)$, and the WS parameter is defined as in (4):(4)$$WS=max\left(\frac{{\bar{W}}^{i}\;\cdot \;{\bar{W}}^{k}}{\left\|{\bar{W}}^{i}\|\;\|{\bar{W}}^{k}\right\|}\right)$$
where ${\bar{W}}^{i}$ and ${\bar{W}}^{k}$ represent the average weight vectors computed across subgroups for the *i*- and *k*-synergy, respectively. The WS value ranges from 0 (i.e., completely dissimilar muscle synergies) to 1 (i.e., completely similar muscle synergies).

#### 2.4.3. Coefficient Similarity

The coefficient similarity (CS) parameter is introduced to select activation coefficients that limit, as much as possible, any redundant information between different muscle synergies. In this case, the correlation of muscle synergies is evaluated between levels *n* and *n* − 1, to check if the splitting of a specific synergy (at level *n* − 1) into two synergies (at level *n*) really provides new information.

For each number *n* of synergies (2 ≤ *n* ≤ 8), and for each synergy *i* (with *i* = 1, …, *n*), the average activation coefficient across subgroups is considered (${\bar{C}}^{i}$), representing the coefficients of a specific synergy over the entire locomotor task. Then, we identify the two (out of *n*) synergies that originated from a specific synergy belonging to the *n*-1 level. These synergies are obtained (except for *n* = 2) by clustering the weights at level *n* into *n*-1 clusters (with the weights of level *n*-1 as centroids), through *k*-means [28]. In this way, the coefficients of the two synergies of interest (${\bar{C}}^{i}$ and ${\bar{C}}^{k}$) will belong to the same cluster (having “forced” *n* elements to cluster into *n*-1 clusters). Finally, “cosine similarity” is introduced to quantify the degree of correlation between the two activation coefficients just identified, and the CS parameter is defined as in (5):(5)$$CS=\frac{{\bar{C}}^{i}\;\cdot \;{\bar{C}}^{k}}{\left\|{\bar{C}}^{i}\|\;\|{\bar{C}}^{k}\right\|}$$

The CS value ranges from 0 (i.e., high information content provided by the new muscle synergy introduced in the level *n*) to 1 (i.e., low information content provided by the new muscle synergy introduced in the level *n*).

#### 2.4.4. ChoOSyn

The ChoOSyn algorithm combines the parameters described above to determine the optimal number of muscle synergies. In particular, for each number of synergies *n* (2 ≤ *n* ≤ 8), we define:(6)$$ChoOSyn_{W}\left(n\right)=WS\left(n\right)+ICV_{W}\left(n\right)$$
(7)$$ChoOSyn_{C}\left(n\right)=CS\left(n\right)+ICV_{C}\left(n\right),$$
for the weights and the coefficients, respectively. The above formulas include the quantification of the synergy similarity through WS and CS to avoid redundant information and the quantification of the synergy consistency across time through ICV to discourage the choice of unstable muscle synergies (see Section 2.4.5—ChoOSyn rules).

Figure 3 shows, as bar diagrams, the values of the $ChoOSyn_{W}$ and $ChoOSyn_{C}$ parameters obtained from the data of two representative (real) subjects. Average bar diagrams of these two parameters were also obtained for the whole simulated and real datasets (and reported in the Results section).

While for the real dataset, we do not know, a priori, the correct number of synergies, this information is known for the simulated dataset. Therefore, through analyzing the bar diagrams of $ChoOSyn_{W}$ and $ChoOSyn_{C}$ extracted from the simulated dataset, it can be seen that, in correspondence with the correct number of synergies (*n* = *n*_c_), there is always a “step” and/or a local minimum. In the following section, this observation will be used to empirically introduce selection rules for obtaining the correct number of synergies. The term “step” refers to a “sharp” increase in the value of the parameter, preceded and followed by “stable” values. The term “local minimum” refers to a situation in which there is an abrupt decrease followed by an abrupt increase in the parameter values [28]. Figure 3 shows examples where both steps and local minima are highlighted (red lines).

To choose the correct number of synergies using both $ChoOSyn_{W}$ and $ChoOSyn_{C}$ parameters we used specific rules detailed below.

#### 2.4.5. ChoOSyn Rules

The presence of a step reveals that muscle synergies maintain an almost steady consistency and similarity as *n* increases (*n* ≤ *n*_c_), but after exceeding *n*_c_ (*n* > *n*_c_) they become highly variable and with redundant information. Instead, the local minimum represents a condition in which there are low values of the ChoOSyn parameters at the level *n*, but if *n* increases or decreases by 1 (*n*−1 and *n*+1 levels), the muscle synergies “get worse”.

To identify steps and local minima, the ChoOSyn algorithm must be able to recognize cases where there is an increase in the value of the parameter from cases where the parameter is almost stable. We introduce the change in the ChoOSyn parameters as *n* increases:(8)ΔChoOSyn(n)=|ChoOSyn(n+1)−ChoOSyn(n)|
(9)with 2≤n≤7

Every variation of the parameter value greater than the average of ΔChoOSyn is defined as an increase, and every variation smaller than $\bar{\Delta ChoOSyn}$ is defined as “stability”. In this way, the algorithm can identify the previously introduced steps and local minima.

If there are multiple steps and local minima in the same bar plot, the algorithm selects only the two highest values of *n* (Figure 3A, left panel).

On this basis, the two parameters $ChoOSyn_{W}$ and $ChoOSyn_{C}$ make two separate selections (Figure 3). Finally, a single optimal value of *n* is chosen as follows:There is at least a common choice in the selection(s) provided by the two parameters (Figure 3A). In this case, the common number of synergies is selected.The two parameters provide a different selection for the number of synergies (Figure 3B). The number is chosen as the one providing the lowest sum of $ChoOSyn_{W}\left(n\right)$ and $ChoOSyn_{C}\left(n\right)$ (i.e., with the lowest similarity and highest consistency).


### 2.5. VAF-Based Methods

As already mentioned in the Introduction, the variance accounted for (VAF) is widely used in the literature to quantify the reconstruction accuracy after the factorization, and it is defined as the uncentered Pearson’s correlation (in percentage) [1,2,18,19,20,21,22,23,24,25]:(10)$$VAF=\left(1-\frac{\sum_{i=1}^{m}{\left(M_{i}-R_{i}\right)}^{2}}{\sum_{i=1}^{m}M_{i}^{2}}\right)\times 100,$$
where M is the matrix before the factorization, R is the reconstructed matrix obtained as the product between W and C, and m is the number of muscles (12 in this work).

We compare the performance of ChoOSyn with the three main VAF-based methods (Figure 4):T-VAF (Threshold VAF) (Figure 4A): this method is the most widely used in the literature [1,2,18,19,20,21,22,23,24,25]. It involves the setting of an arbitrary threshold and the subsequent choice of the first number of synergies with VAF above the threshold. The threshold is commonly set at 90% and less frequently at 95%: therefore, we chose to test both 90% and 95% thresholds.E-VAF (Elbow VAF) [11] (Figure 4B): this method requires finding the “elbow” of the VAF curve, i.e., the highest curvature point. It is the only VAF-based method that does not use arbitrary thresholds.P-VAF (Plateau VAF) [40] (Figure 4C): this method requires finding the point beyond which the VAF curve reaches a plateau. It uses an arbitrary threshold: the mean-square error obtained by fitting the VAF-curve through a straight line must be smaller than 10^−2^. Cheung et al. [40] used a threshold equal to 10^−5^, but in our simulated dataset 10^−2^ provided the best performance. The first point satisfying this condition is chosen.

### 2.6. Performance Evaluation

The performances of the ChoOSyn algorithm were compared to those of T-VAF, E-VAF, and P-VAF methods by means of the fraction of correct classifications, mean error (ME), and root mean square error (RMSE), both for the simulated and real datasets. RMSE is used to quantify how far a given method deviates from the correct number of synergies, while ME provides information about the sign, to know whether the method goes wrong by defect or excess. ME and RMSE are defined as follows:(11)$$ME=\frac{\sum_{i=1}^{NS}\left(n_{i}-n_{c,i}\right)}{NS}$$
(12)$$RMSE=\sqrt{\frac{\sum_{i=1}^{NS}{\left(n_{i}-n_{c,i}\right)}^{2}}{NS}}$$
where n is the number of synergies identified by a method, *n*_c_ is the number of correct synergies, and NS is the total number of subjects in the dataset.

To know which number of synergies should be considered as correct (*n*_c_) in the real dataset, we developed a “ground truth” using the judgment of two expert operators. Their judgment was performed blind to the details of the ChoOSyn algorithm as well as to the results of the various methods tested. For each real subject, they analyzed the muscle synergy plots considering different numbers of muscle synergies *n* and they chose—separately—the number they considered as correct, based on their knowledge of motor control strategies, muscle synergy analysis, and gait biomechanics. It should be noted that expert judgment is subjective, at least to some extent. Cohen’s kappa statistic [41] was used to compute the degree of agreement between the raters. In case of disagreement, the two expert operators discussed the discordant cases to achieve a common ground truth. For the simulated dataset, its own nature guarantees its objectivity, knowing a priori the correct number of muscle synergies.

## 3. Results

### 3.1. Simulated Data

Figure 5A–C show the two ChoOSyn parameters extracted from the simulated dataset. More specifically, data obtained simulating *n* = 4, *n* = 5, and *n* = 6 muscle synergies are displayed. The bar plots show a marked “step”, in correspondence to the correct number of synergies. This is the main feature that allows the algorithm to identify the optimal number of synergies without thresholds. In addition, in some cases, the plots present a local minimum just before the step.

The final results obtained applying the ChoOSyn rules are reported in Table 1. The row “no noise” shows the performance of the different methods tested without additive noise. The results obtained considering increasing levels of additive noise are also reported to evaluate the robustness of the methods at different SNR values.

Overall, T-VAF methods fail to identify the correct numbers of synergies, while E-VAF, P-VAF, and ChoOSyn show comparable performances (except for SNR = 15 dB).

The synthetic signals are less complex to factorize, and, hence, the reconstruction accuracy (VAF) shows higher values already at lower numbers of synergies. Indeed, T-VAF identifies as the optimal number of synergies 1 to 4 units lower than the correct one. T-VAF also shows the highest ME and RMSE values.

The performance of T-VAF (95%) decreases with increasing additive noise, while, considering E-VAF, P-VAF, and ChoOSyn, the performance degradation is notable only in the worst condition (at 15 dB). For the ChoOSyn algorithm, this result was also predictable from the bar plots of Figure 5A–C. There is a marked step in the case without additive noise (purple bar) and, to a lesser extent, with high SNR values (red, orange, and yellow color bars), while the step becomes markedly shorter for the green bars representing SNR = 15 dB.

### 3.2. Real Data

Figure 5D–F show the two ChoOSyn parameters extracted from the real dataset. Overall, trends observed in the simulated dataset (Figure 5A–C) are also present in the real dataset (Figure 5D–F). In the latter case, we used the expert ground-truth to divide the population into subjects that express 4, 5, and 6 muscle synergies. The inter-rater agreement, computed by means of Cohen’s kappa, was equal to 0.5, suggesting a moderate agreement between the two expert operators.

Considering the real dataset (Table 2), ChoOSyn achieved the best performance, with 17 out of 20 correct classifications and the lowest ME and RMSE. A slightly worse performance was observed for E-VAF, which obtained 12 out of 20 correct classifications. T-VAF and P-VAF achieved the worst performances.

## 4. Discussion

Choosing the correct number of muscle synergies that control a motor task is fundamental to understand how the CNS drives the muscles. However, the method employed for selecting the correct number of synergies plays a critical role. The number of synergies characterizing a given activity, e.g., locomotion, varies within and across studies, even for unimpaired individuals [28]. There is a lack of standardized methods for the precise identification of the number of synergies, making comparisons across studies and cohorts difficult.

The method currently accepted and used by the vast majority of researchers is based on VAF (variance accounted for) [1,2,18,19,20,21,22,23,24,25], which quantifies the reconstruction accuracy, i.e., how faithfully the muscle synergies represent the signals collected from the muscles. Among the various VAF criteria applied to select the optimal number of synergies, the threshold-VAF (T-VAF) is the most widely adopted, although, it relies on the definition of a fixed threshold T (i.e., T = 90%, Figure 4A). The first number of synergies that produces a VAF value equal to or greater than the threshold is selected as optimal. In the literature, the threshold T is commonly set at 90% [1,2,4,7,18,20,21,22,23,24], and less frequently at 95% [19,25].

The T-VAF method is very simple to implement, but, on the other hand, the presence of a fixed threshold is a well-known issue. First, the threshold is set arbitrarily and different research groups may use different T-values: globally, there is a lack of clear criteria to choose a specific value with respect to another. Second, a small variation in the T-value could also significantly change the results. Therefore, it would be necessary to test the robustness of the threshold itself.

In this work, we proposed and validated a method to choose the optimal number of muscle synergies (ChoOSyn), which is independent of the definition of arbitrary thresholds. ChoOSyn is an alternative to VAF and relies on two parameters directly estimated from data: consistency and similarity of muscle synergies.

Other research groups have introduced alternatives to VAF cutoff criteria. More specifically, Cheung et al. [16] proposed a statistical approach based on real and unstructured sEMG signals, generated by randomly shuffling the original sEMG signals across time and muscles, to select the correct number of muscle synergies in a more interpretable way with respect to standard threshold-based approaches. Ref. [28], instead, introduced intra-class and between-level correlation coefficients to discriminate “reliable” from “unreliable” synergies. Their approach was based on *k*-means clustering and was tested on 9 healthy subjects, considering eight leg muscles during treadmill walking. Delis et al. [29], [30] developed a more “physiological” approach, introducing a task decoding-based metric during an arm pointing task.

The approach proposed in this work was validated both on a simulated and on a real dataset, considering an overground walking task. The performance of ChoOSyn was directly compared against VAF-based methods, in terms of the fraction of correct classification, mean error (ME), and root mean square error (RMSE).

Analyzing the simulated dataset, we found that ChoOSyn correctly identified the number of synergies in almost all cases with very small errors. E-VAF and P-VAF methods showed overall performances similar to ChoOSyn. On the contrary, the T-VAF method fails to identify the correct number of muscle synergies. This is probably due to the nature of the dataset: the simulated signals are less complex to factorize, and the VAF assumes higher values already at small numbers of synergies. Indeed, the ME values show that the T-VAF always goes wrong by defect.

When tested on the real dataset, ChoOSyn achieved the best performance, with a marked difference compared to the other methods. The ME and RMSE values of ChoOSyn are also much lower than those of the other VAF-based methods. The worst performing methods were T-VAF and P-VAF, which are based on arbitrary thresholds, further highlighting the problem mentioned above. E-VAF, which does not require thresholds, has the best performance of the VAF-based methods.

Therefore, we proved that ChoOSyn shows equal (simulated dataset) or even higher (real dataset) performance in the correct identification of the number of synergies with respect to the methods currently available in the literature. Indeed, the misclassifications are limited and the number of synergies obtained is close to the correct number, with ME and RMSE values comparable (simulated dataset) or smaller (real dataset) than those obtained with VAF-based methods. Moreover, ChoOSyn operates without thresholds.

The number of muscle synergies can also be strongly influenced by other steps of the muscle synergy extraction process, such as the sEMG pre-processing (e.g., low-pass filtering techniques) [42] and the number and choice of muscles acquired [18]. However, the focus of this contribution is on developing an approach that can be applied after a factorization algorithm, to select the correct number of muscle synergies, and not on evaluating the effect of different pre-processing techniques on the identification of the synergy number. We demonstrated that the ChoOSyn algorithm is more reliable than VAF-based methods. This suggests that the two newly introduced ChoOSyn parameters and the concepts behind them are relevant. It is desirable to obtain a high consistency of muscle synergies over the motor task duration, and low intra-level similarity between synergies (avoiding redundant information). Following these guidelines facilitates the proper selection of the correct number of synergies. We found that a method based on these concepts is more discriminative than the reconstruction accuracy (at the base of VAF-methods) in the search for the correct number of muscle synergies.

The proposed method was tested on 20 healthy subjects. It would be interesting to test.

ChoOSyn both on a larger population and on different cohorts (for age or pathological condition). The need for long-lasting sEMG acquisitions to properly select the number of muscle synergies does not limit the feasibility and applicability of the proposed approach to pathological populations. Indeed, gait analysis is commonly used only in those patients that can independently walk, for at least some minutes, without external supports or walking aids. In the past, several studies demonstrated the feasibility of long-lasting gait data acquisition in patients suffering from different neurological conditions, such as normal pressure hydrocephalus [43], mild ataxia [44], and cerebral palsy [45]. Future work should focus on providing algorithm validation for patients affected by neurological disorders, such as patients affected by Parkinson’s disease or stroke survivors, by also increasing the number of expert operators for the “ground truth” definition. Moreover, the dataset used includes signals acquired from the lower limb and the trunk while walking. However, the ChoOSyn method is not necessarily associated with the specific motor task considered and can be generalized to signals acquired during a different motor task or from different muscles. Indeed, since the proposed approach does not rely on arbitrary thresholds or task-dependent rules, it can be potentially extended to other cyclic motor tasks, such as running or cycling.

## 5. Conclusions

We described and validated an algorithm (ChoOSyn) to select the optimal number of synergies expressed during gait, which overcomes the limitations of VAF thresholding methods. The proposed approach may support the standardization of reports, in motor control studies, among different research laboratories. Moreover, ChoOSyn may be applied to different repetitive motor tasks (reaching movements of the upper limbs, etc.) without any specific need for adaptation to the motor task considered.

## Figures and Tables

**Figure 1 sensors-21-03311-f001:**
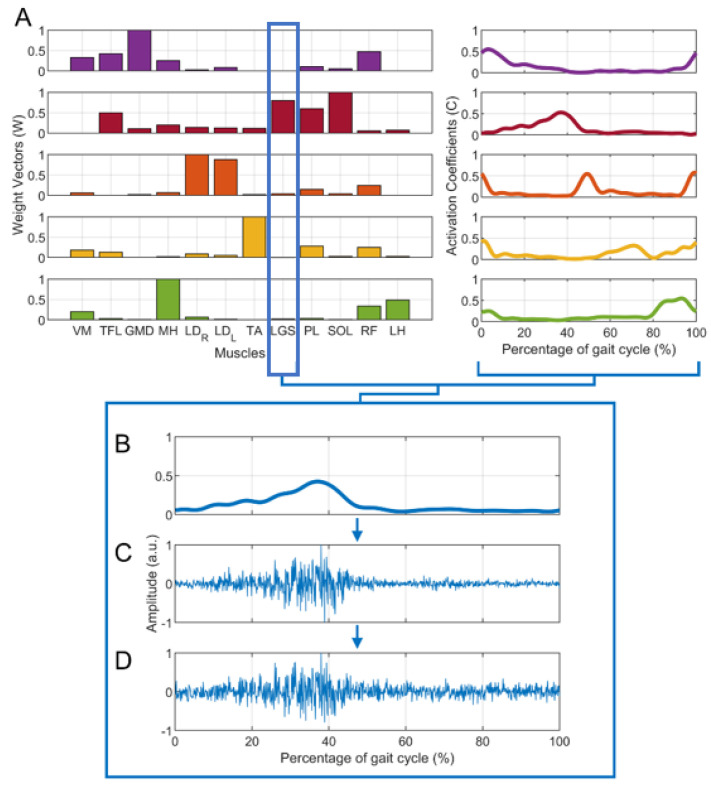
Example of the generation of a simulated sEMG signal for the lateral gastrocnemius (LGS) muscle: the first step is (**A**) the extraction of muscle synergies (W and  C ) from the real data of a representative subject with 5 muscle synergies, the second is (**B**) the reconstruction of the LGS envelope (obtained as  W_LGS_ * C ). Then, (**C**) a simulated sEMG signal without additive noise is generated. Finally, noise is added to the previous signals. An example of a simulated sEMG signal with SNR = 20 dB is shown in (**D**).

**Figure 2 sensors-21-03311-f002:**
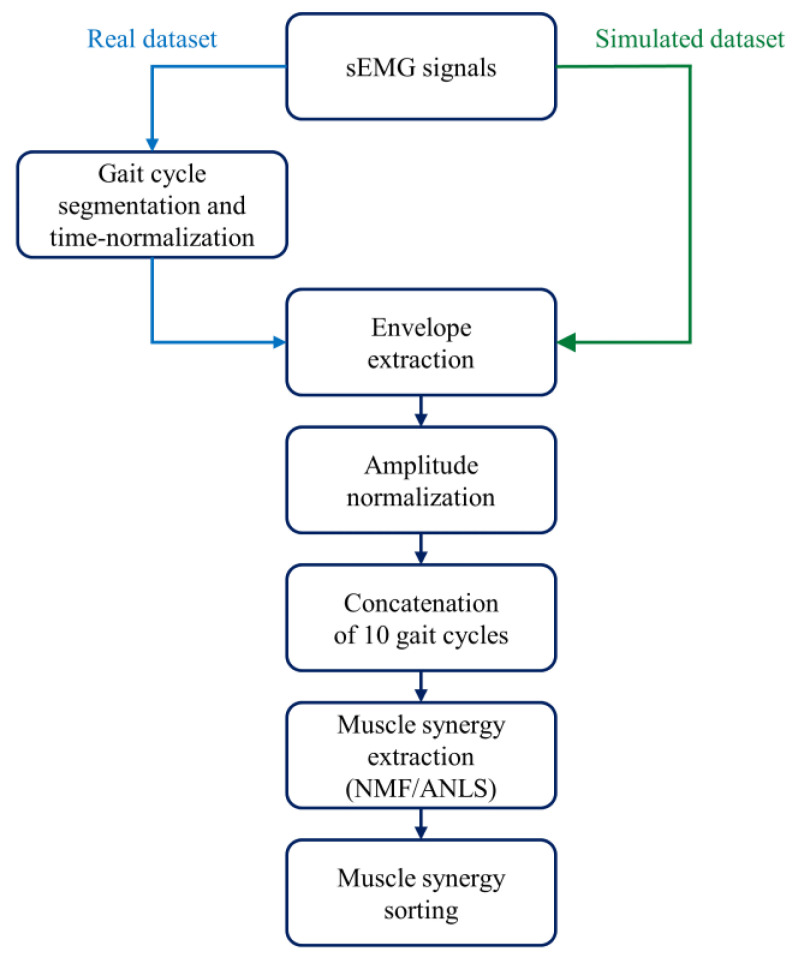
Data processing: steps to extract muscle synergies from the sEMG signals.

**Figure 3 sensors-21-03311-f003:**
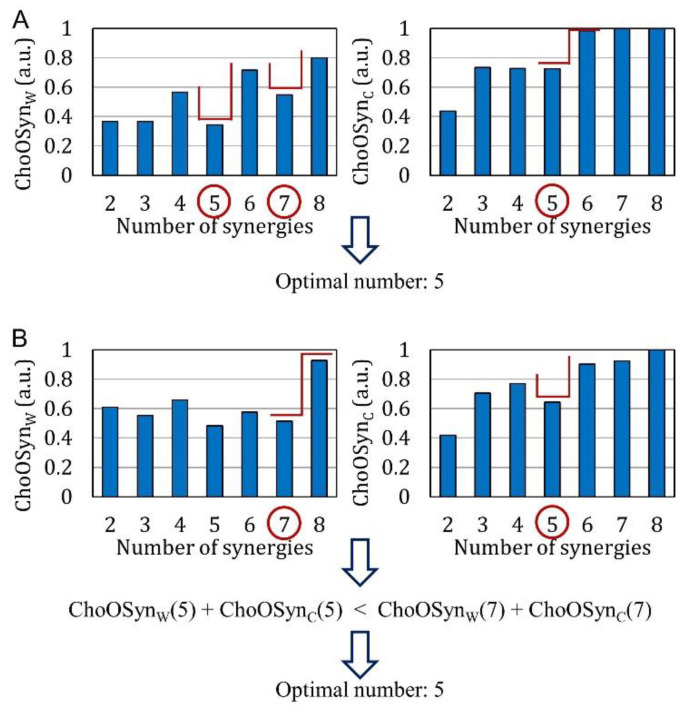
Examples of $ChoOSyn_{W}$ and $ChoOSyn_{C}$ values calculated on muscle synergies extracted from the data of two representative real subjects. “Steps” and local minima are highlighted by red segments. These examples show how the optimal number of synergies is chosen when the outputs of the two parameters are (**A**) the same or (**B**) different.

**Figure 4 sensors-21-03311-f004:**
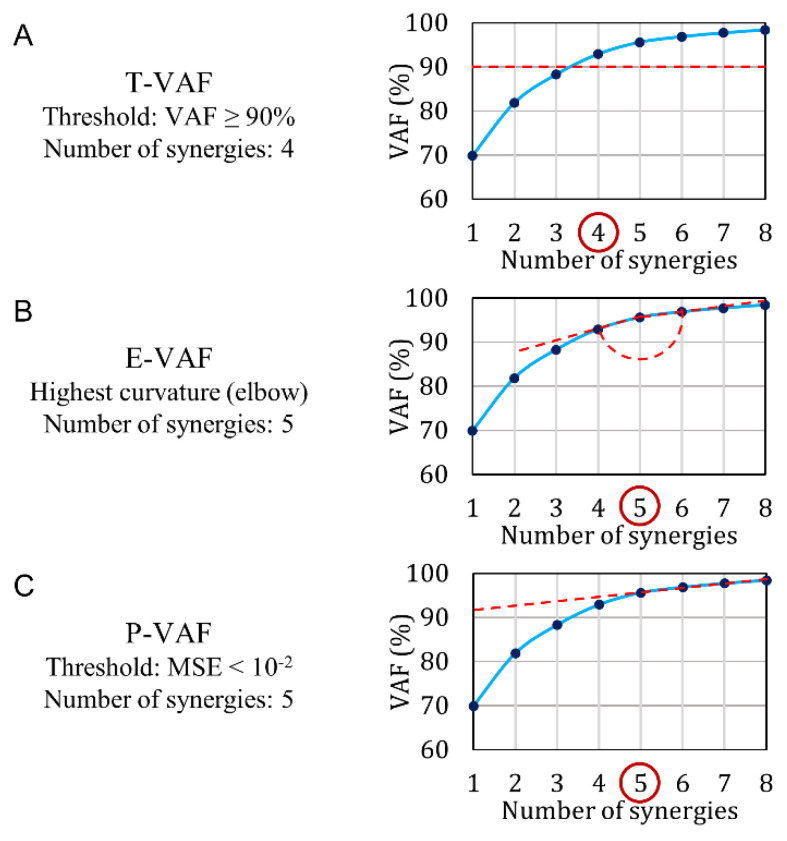
VAF-based methods used in the literature: (**A**) T-VAF, the most used method, (**B**) E-VAF, based on the curvature, and (**C**) P-VAF, based on the plateau. The reported VAF curve is calculated from the data of a real representative subject (the same for Figure 3 and Figure 4).

**Figure 5 sensors-21-03311-f005:**
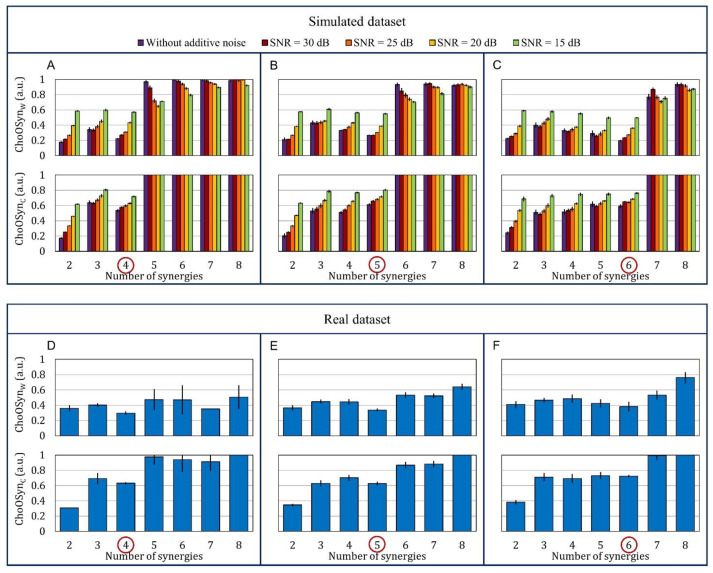
Bar plots representing the mean ± standard error of the two parameters $ChoOSyn_{W}$ and $ChoOSyn_{C}$. (**A**–**C**) Upper plots: simulated dataset. Each bar represents a different noise condition. The dataset is divided into three subsets with (**A**) 4, (**B**) 5, and (**C**) 6 muscle synergies, respectively. (**D**–**F**) Bottom plots: real dataset. It is divided into subjects that express (**D**) 4, (**E**) 5, and (**F**) 6 muscle synergies, respectively. We used the ground truth to divide the real dataset into three subsets.

**Table 1 sensors-21-03311-t001:** Simulated dataset—performance of the different methods in terms of the fraction of correctly classified, mean error (ME), and root-mean-squared error (RMSE).

**Fraction of Correctly** **Classified**	**T-VAF** **(90%)**	**T-VAF** **(95%)**	**E-VAF**	**P-VAF**	**ChoOSyn**
No noise	2/75	36/75	75/75	75/75	73/75
SNR = 30 dB	0/75	24/75	75/75	75/75	73/75
SNR = 25 dB	0/75	18/75	75/75	75/75	74/75
SNR = 20 dB	0/75	9/75	74/75	72/75	72/75
SNR = 15 dB	0/75	0/75	65/75	73/75	63/75
**ME ^1^**	**T-VAF** **(90%)**	**T-VAF** **(95%)**	**E-VAF**	**P-VAF**	**ChoOSyn**
No noise	−1.29	−0.52	0.00	0.00	−0.03
SNR = 30 dB	−1.48	−0.68	0.00	0.00	−0.03
SNR = 25 dB	−1.57	−0.79	0.00	0.00	−0.01
SNR = 20 dB	−2.11	−1.04	0.01	0.04	−0.05
SNR = 15 dB	−3.07	−1.93	−0.15	0.03	0.04
**RMSE ^1^**	**T-VAF** **(90%)**	**T-VAF** **(95%)**	**E-VAF**	**P-VAF**	**ChoOSyn**
No noise	1.39	0.72	0.00	0.00	0.16
SNR = 30 dB	1.56	0.82	0.00	0.00	0.16
SNR = 25 dB	1.65	0.92	0.00	0.00	0.12
SNR = 20 dB	2.19	1.17	0.12	0.20	0.28
SNR = 15 dB	3.15	2.00	0.53	0.16	0.53

^1^ Unit of measure of ME and RMSE: number of synergies.

**Table 2 sensors-21-03311-t002:** Real dataset—Performance of the different methods as the fraction of correctly classified, mean error (ME), and root-mean-squared error (RMSE).

	T-VAF (90%)	T-VAF (95%)	E-VAF	P-VAF	ChoOSyn
Fraction of correctly classified	8/20	7/20	12/20	6/20	17/20
ME ^1^	−0.90	0.55	0.70	0.90	0.20
RMSE ^1^	1.30	0.98	1.18	1.18	0.55

^1^ Unit of measure of ME and RMSE: number of synergies.

## Data Availability

Data presented in this study are available on request from the corresponding author.

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
