# Peer review of "An Algorithm for Choosing the Optimal Number of Muscle Synergies during Walking"

_sensors, 2021, doi:10.3390/s21103311_

Round 1

Reviewer 1 Report

In this study, R. Ballarini and colleagues propose an algorithm to select the optimal number of muscle synergies to be extracted from multi-muscle EMG recordings. The method is based on the characteristics of the synergies, in terms of consistency and similarity. The method is tested on a set of synthetic EMG signals and a dataset of experimentally obtained EMG signals, and its performances are compared with the most used algorithms in literature. The authors show that, while the performance on simulated data is comparable with that of traditional algorithms, the proposed method performs better when analyzing experimental EMG signals.

This study moves from the lack of standardization regarding the model order selection in muscle synergy analysis, and the manuscript is well contextualized in this sense. The methodology takes into account the VAF-based criteria using a hard threshold to select the number of synergies. The methods are well described allowing a good reproducibility of the results. The manuscript is in general well structured, and the main message coming from the proposed method is adequately conveyed. However, I think that reducing all the analysis and the comparison to VAF-based algorithms with a hard threshold does not include some important methods, used in previous studies and less dependent from a subjective choice. I also have some doubts regarding the generation of the simulated (pseudo-real) dataset and on the management of the real dataset. Moreover, I think that some important assumptions related to the development and practical implementation of the ChoOSyn method need to be better justified.

A list of detailed points is following:

  1. The significance of the number of synergies is not taken into account in the introduction. I suggest inserting some references related to clinical and motor learning studies, highlighting how the number of synergies is a fundamental parameter, well corelated to the motor performance of patients or subjects [1-4]
  2. Some previously used methods for selecting the number of synergies are not explored. Previous works not only proposed a subjective hard threshold, but proposed some statistical method based on the properties of the VAF curve when surrogate data are used, as in [5] (corresponding to the present reference [37] in the manuscript). If not directly analyzed, these methods should at least be mentioned for completeness.
  3. The construction of the simulated data is well described. However, they are not synthetic data, but pseudo-real data. I identify two problems out of this consideration: first, the real number of synergies comes from an a priori assumption on the data, I suppose by using the traditional algorithms. Secondly, using this pseudo-real dataset, the ChoOSyn algorithm is only tested on number of synergies equal to 4,5 and 6, so that we don’t know if the algorithm performs properly on a lower number (as it happens, for example, in stroke patients, [1-3]). A number of synergies higher than 6 has been never encountered in any task to my knowledge, so I agree with using 6 as a upper limit.
  4. Lines 115-121. Actually, this procedure comes from an envelope that already contains noise from the previous envelope calculation (that is intrinsically present in the extracted synergy activation coefficients). So stating that “no noise is added at this stage” does not imply that no noise is present, since the initial SNR strongly depends on the SNR of the envelope itself, and it is transferred to the modulated WGN (as evident in figure 1-C)
  5. Lines 170-174: these requirements appear to be subjective, as there is no convincing justification for these characteristics. While the consistency across subgroups is reasonable, given that the task is not changing in terms of spatiotemporal control (I assume that speed and stride length were quite consistent along the trial and across subgroups), I don’t fully understand the rationale for the low similarity across synergies. Two synergies may be similar for biomechanical reasons, and it can happen that one or two muscles are shared across different muscle synergies, depending on the set of recorded muscles. Given the short explanation at lines 176-177, does the method work only on locomotion EMG data? In this case, I would suggest to change the title of the paper, including this latter aspect (e.g. “An algorithm for choosing the optimal number of muscle synergies during walking/ in gait/in locomotor tasks”). However, a more elaborated explanation for this choice is required.
  6. Lines 134-135: only on real data. Even though it is redundant, specify that this is done only on real data since simulated data actually come from the same real data. The present reading seems to suggest that only real data are time normalized.
  7. Paragraph 2.4.3 - Coefficient similarity: while I think this is an interesting marker of information redundancy when increasing the number of synergies from n to n+1, I wonder whether this value can be influenced by the low-pass cut-off frequency for calculating the amplitude envelope (low pass effect that is even highlighted by calculating the average activation coefficient).
  8. Paragraph 2.4.4 – ChoOSyn: the description related to the presence of a step or “sometimes” a local minimum appears subjective and not fully justified. I think some additional explanation is needed, to prove that this peculiarity is related to optimality. For example, the rules described at lines 274-281 appear arbitrary.
  9. I have a general consideration related to the practical use of ChoOSyn. Based on the description of the algorithm, the recorded data need to be divided into subgroups to select the optimal number of synergies. This implies that, for the algorithm to be properly working, a long recording with a high number of gait cycles is required. This constitutes a big limitation to the use of the proposed technique, as in many situations, especially those experimental scenarios involving pathological subjects, a high number of gait cycles might not be available.
  10. Lines 299-304: actually in [37] a different method is used. The one described in the present manuscript is the one used in [36].
  11. Lines 314-324: I do not fully agree with this approach. Given the obtained results, that elect ChoOSyn as the best performing algorithm only on the real dataset, the choice made by the two expert operators strongly influence the results of this study. I suggest using an alternative method, not dependent on the operator selection. You can either strongly increase the amount of independent operators, or select the best performing algorithm on the simulated data as the ground truth.
  12. Line 252: there is typo -  repetition of [24]
  13. Lines 420-425 in the discussion. This is quite an obvious sentence. Indeed, the simulated data should help you selecting the correct number of synergies in the real data, by using the algorithm that performs better on the simulated data. This is even more evident as the simulated data are pseudo-real data.

References:

[1] Ambrosini, E., De Marchis, C., Pedrocchi, A., Ferrigno, G., Monticone, M., Schmid, M., ... & Ferrante, S. (2016). Neuro-mechanics of recumbent leg cycling in post-acute stroke patients. Annals of biomedical engineering, 44(11), 3238-3251.

[2] Clark, D. J., Ting, L. H., Zajac, F. E., Neptune, R. R., & Kautz, S. A. (2010). Merging of healthy motor modules predicts reduced locomotor performance and muscle coordination complexity post-stroke. Journal of neurophysiology, 103(2), 844-857.

[3] Routson, R. L., Clark, D. J., Bowden, M. G., Kautz, S. A., & Neptune, R. R. (2013). The influence of locomotor rehabilitation on module quality and post-stroke hemiparetic walking performance. Gait & posture, 38(3), 511-517.

[4] Sawers, A., Allen, J. L., & Ting, L. H. (2015). Long-term training modifies the modular structure and organization of walking balance control. Journal of Neurophysiology, 114(6), 3359-3373.

[5] Cheung, V. C., Piron, L., Agostini, M., Silvoni, S., Turolla, A., & Bizzi, E. (2009). Stability of muscle synergies for voluntary actions after cortical stroke in humans. Proceedings of the National Academy of Sciences, 106(46), 19563-19568.

Reviewer 2 Report

Thank you for the opportunity to review the manuscript titled ‘An algorithm for choosing the optimal number of muscle synergies’. The authors have created an algorithm to create and validate an algorithm that is designed to identify the ‘optimal’ number of synergies. They used a real and synthetic data set to validate their optimal identification algorithm. The rational for the algorithm is that the VAF method is a subjective arbitrary value that is selected by researchers but is inconsistent throughout the literature.

The manuscript needs to be grammatically improved. The paragraph structures and use of commas need to be addressed and improved.

Main Questions:

1: The authors do not test their algorithm against VAF studies that use other criterion (i.e. addition of another synergy does not result in noticeable increase in the VAF value). Did the authors tests against identification methods that use more rule-based identifications in addition to the VAF value?

2: The authors use the ‘step’ method to identify optimal number of synergies. This method is discussed to provide a non ‘arbitrary’ method of identification, but they do not provide a quantifiable increase in the ChoOSyn variables that satisfied the step and local min components. Did the authors in the development of the algorithm create a quantifiable delta that identifies the location that marks a ‘step’?

Introduction:

Line 50: change ‘methods’ to ‘methodology’

Line 51: reword ‘vast majority of works in the literature’.

Methods & Materials:

Line 72-77: Not sure this paragraph is needed.

Section 2.3: Provide rational for the processing used. Authors used an 8th order butterworth and then again a 5th order to obtain the sEMG envelops.

For each of the sections describing the calculation variables (2.4.1, 2.4.2, and 2.4.3) use the same format for describing what the 0-1 score means. Lines 195-197, Lines 209-211, & line 230-231 should all use the same format. Just choose a style and use it for all three.

Line 245: Add a reference to ‘section 2.4.5’ after ‘the correct number of synergies’ to orient the reader where they will find the description of how this was identified.

Lines 314-320: Identify the criterion the ‘experts’ used to identify the number of muscle synergies for the ‘ground truth’ identification. Did they use the same criterion or did they use different criteria? This is also relevant to describe why they only had a moderate agreement between themselves (line 360-362).

Discussion

Line 373: Consider rewording ‘As already pointed out in Ref[24]. If you just state what was pointed out with a reference at the end of the statement, the reader understands where the statement comes from.

Line443-445: Please develop this claim more. Why do the authors believe this algorithm has the capacity to be generalized to other motor tasks?

There are errors in the reference list, please review and fix.

Round 2

Reviewer 1 Report

I thank the authors for their good work of revision and for the clear response to my previously raised issues. I have no further concern with the manuscript.